# Effectiveness of Integrated Education to Reduce Postoperative Nausea, Vomiting, and Dizziness after Abdominal Surgery under General Anesthesia

**DOI:** 10.3390/ijerph18116124

**Published:** 2021-06-06

**Authors:** Yoonhee Seok, Eunyoung E. Suh, Soo-Young Yu, JeongYun Park, Hyunjin Park, Eunsil Lee

**Affiliations:** 1Center for Human-Caring Nurse Leaders for the Future by Brain Korea 21 (BK 21) Four Project, College of Nursing, Seoul National University, Seoul 03080, Korea; uri303@snu.ac.kr; 2Center for Human-Caring Nurse Leaders for the Future by Brain Korea 21 (BK 21) Four Project, Research Institute of Nursing Science, College of Nursing, Seoul National University, Seoul 03080, Korea; 3Department of Nursing, Chungbuk National University, Cheongju 28644, Korea; sooyoung0809@gmail.com; 4Department of Clinical Nursing, Ulsan University, Seoul 05505, Korea; pjyun@ulsan.ac.kr; 5Department of Nursing, Asan Medical Center, Seoul 05505, Korea; matmeet1119@naver.com (H.P.); jupiter0210@hanmail.net (E.L.)

**Keywords:** general surgery, postoperative nausea and vomiting, dizziness, education

## Abstract

This study presents an anticipatory integrated education program for nausea, vomiting, and dizziness prevention (anti-NVD education program) for patients undergoing abdominal surgery under general anesthesia. The anti-NVD education program for nephrectomy patients consisted of the following: the causes of postoperative nausea, vomiting, and dizziness; effective deep breathing and how to use an inspirometer; postoperative nausea and vomiting; effective methods of patient-controlled analgesia; and the stepwise standing up method to prevent dizziness. A study was conducted among 79 adults (experimental group: *n* = 40, control group: *n* = 39). The degree of nausea and dizziness was measured using a numerical rating scale (NRS), and vomiting and the frequency of antiemetic use were measured in terms of the number of patients. The experimental group, which received the anti-NVD education, showed remarkably lower levels of nausea (*p* = 0.013) and dizziness (*p* < 0.001) than the control group. The frequency of antiemetic use 48 hours after surgery was significantly lower in the experimental group (*p* = 0.03). This study proved the efficacy of the anti-NVD education program for reducing postoperative nausea and dizziness. This program can be used as a noninvasive nursing intervention to prevent nausea, vomiting, and dizziness among patients undergoing abdominal surgery.

## 1. Introduction

Due to developments in modern medicine, surgery is increasingly frequently performed for diagnostic and treatment purposes. Among patients who visit hospitals in South Korea (hereafter, Korea), the number of patients who receive surgical treatment is increasing each year [1]. The total number of surgical procedures in Korea increased by an annual average of 2.3% within 5 years, from 1.17 million cases in 2014 to 1.87 million cases in 2018 [2]. The number of patients undergoing abdominal surgery under general anesthesia is gradually rising, leading to an increased demand for nursing interventions to reduce patients’ postoperative discomfort [3,4], which manifests as pain, nausea, vomiting, dizziness, drowsiness, and headache [5]. Previous research on nursing interventions after surgery in Korea has been unequally distributed among pharmaceutical methods and non-pharmaceutical methods (relaxation therapy, music therapy, aromatherapy, Nei-Guan acupressure) for pain reduction [6,7,8]. However, research on postoperative dizziness has been limited to surgery for specific conditions, such as chronic otitis media and brain tumor [9], and dizziness in patients undergoing abdominal surgery has not received significant recognition as a nursing problem requiring nursing interventions, unlike pain, nausea, and vomiting, which have been addressed relatively frequently [10].

Although postoperative dizziness is not a severe, life-threatening complication, it does have major implications for patients undergoing abdominal surgery under general anesthesia. Early ambulation after abdominal surgery is essential for the rapid recovery of bowel movements [11]. Dizziness occurring when a patient first gets up from bed after surgery causes problems related to the patient’s recovery and safety. According to previous studies, 17% of falling accidents in hospitals occur due to dizziness, and 5% of them cause external injuries, which may cause serious complications [12]. The seriousness of dizziness has been under-acknowledged; therefore, dizziness and postoperative nausea and vomiting are a nursing problem that requires nursing interventions.

Research on the prevalence of postoperative dizziness has shown tremendous variation, between 20% and 93%, and relatively few studies have analyzed the intensity of dizziness [13]. A study found that the intensity of patients’ dizziness varied across a broad spectrum, making it difficult to define the intensity of dizziness rigorously [14]. Research on postoperative dizziness is insufficient, as only three studies have identified factors related to dizziness. Possible causes of dizziness in postoperative patients include remaining anesthetic, minor anxiety, and the reduction of blood flow to the brain when a patient suddenly gets up after lying down for a long time, and associations have also been found with female sex, surgery on the inferior rectus, and surgery on the inferior oblique [15,16,17].

Sudden movements, which are a cause of dizziness, are known to be related to nausea and vomiting. As postoperative patients were moved from the recovery room to the ward, the symptoms of nausea and vomiting increased to a remarkable extent [18]. Starting to move after surgery caused nausea and vomiting, which were correlated with their motions when they moved or were transferred [19]. To reduce postoperative patients’ nausea, vomiting, and dizziness, patients’ motions should be minimized, and sudden motions should be avoided during transfer [20]. Dizziness caused by sudden motion is related to blood flow. When a person lies down and then stands up on both feet, 500−1000 mL of blood descends due to gravity, and then remains in the venous blood storage system in the abdomen under the diaphragm, the pelvis, and the lower limbs. The return of venous blood decreases due to blood flow, and ventricular enlargement decreases. Therefore, cardiac output decreases, blood pressure drops, and a person may feel dizziness for several seconds because of reduced blood flow to the brain [17]. Intraoperative blood loss, which may reach 50−1500 mL and 50−800 mL in open and laparoscopic nephrectomy, respectively, makes venous return difficult upon standing up after surgery [21]. Therefore, to prevent dizziness, it is essential to educate patients on the need to sit or stand up slowly from a lying-down position.

Among the factors associated with dizziness, remaining anesthetic, the patient’s sex, and surgery type cannot be changed by a nursing intervention. However, nurses can intervene to minimize sudden motions. Therefore, an education program for nausea, vomiting, and dizziness prevention (anti-NVD education program) to prevent sudden motions before surgery is necessary to prevent postoperative nausea, vomiting, and dizziness effectively. However, no research on nursing interventions for dizziness in patients undergoing abdominal surgery has yet been conducted. Since the cause of dizziness in the postoperative condition is related to sudden movement, this is a part that can be sufficiently mediated by nurses’ stepwise standing up method education.

Nursing education interventions that prevent the discomfort associated with surgery and help patients effectively adapt to the postoperative situation are cost-effective and can be independently implemented by nurses who observe the patient’s condition in real time from the patient’s side. In a study on same-day surgery patients, insufficient information provided before surgery caused difficulties in coping with the situation and caused anxiety. In addition, preoperative pain and nausea management education were found to be effective in patients’ early discharge and pain reduction [22]. Although not surgery-related nausea and vomiting, the experimental group that received self-management behavioral education on chemotherapy-related nausea, vomiting, and fatigue for breast cancer patients showed lowered anxiety and improved symptoms, such as nausea and vomiting [23].

For a postoperative patient’s speedy recovery, treatment to minimize postoperative side effects is necessary, but nursing education to prevent side effects is also vital [24]. The key to anticipatory care is to set goals to solve problems that can occur in treatment and try an anticipatory approach early [25]. Anticipatory care arose from the concept that preventing problems is more effective than treating problems that have already occurred [26]. As an example, applying a guideline for preventing sleep problems led to a reduction in sleep disorders in newborn babies, a lower prevalence of parent-reported night waking, and less fatigue and depression among mothers and their partners [27].

This study presents the development and implementation of an anti-NVD education program to solve problems such as nausea, vomiting, and dizziness that may occur after surgery. This research aimed to examine the effects of this program on postoperative nausea, vomiting, and dizziness among the patients undergoing abdominal surgery.

## 2. Materials and Methods

### 2.1. Study Design

This study targeted hospitalized patients who underwent nephrectomy at the urology department of a hospital and were offered an anti-NVD education program. A quasi-experimental (nonincidental control group-time difference) design was used to compare the effects of the anti-NVD education program between the experimental and control groups.

### 2.2. Participants

The participants were adult patients who underwent nephrectomy under general anesthesia at a general hospital in Seoul, Korea. The criteria for selecting participants were as follows: (1) patients undergoing nephrectomy under general anesthesia who received nephrectomy; (2) patients to whom intravenous patent-controlled analgesia (IV-PCA) was applied; (3) patients with an American Society of Anesthesiologists (ASA) physical status of class 1 or 2; (4) patients who received general anesthesia for over an hour; and (5) patients who could understand questions and could communicate. The exclusion criterion was discontinuation of IV-PCA use during the study.

To calculate the sample size, the effect size was set to 0.70 based on the educational effects reported Jeon et al. [28]. Using the G*Power 3.1.3 program (https://www.psychologie.hhu.de/arbeitsgruppen/allgemeine-psychologie-und-arbeitspsychologie/gpower accessed on 23 August 2013), the required number of participants was calculated for a significance level of 0.05 and a power of 0.90, using the t-test. The resulting sample size was 36 in each group. Considering a possible dropout rate 10%, a total of 80 patients, divided into two groups (*n* = 40 each), were selected [29].

### 2.3. Process of Developing the Anticipatory Integrated Education Program

An anti-NVD education program was developed to reduce nausea, vomiting, and dizziness in nephrectomy patients based on the principle of anticipatory care. In detail, the anti-NVD education program consisted of the following: causes of postoperative nausea, vomiting, and dizziness; effective deep breathing and how to use an inspirometer; effective use of IV-PCA; the stepwise standing up method to prevent dizziness; and the importance of early ambulation to prevent paralytic ileus (Table 1). Education on the causes of nausea and vomiting can improve patients’ knowledge and attitudes regarding coping with side effects [30]. Deep breathing and inspirometer education can help prevent pulmonary complications, promote exhalation of the remaining anesthetic inhalant after surgery [31], and prevent dizziness due to the inhaled anesthetic.

Education affects pain reduction at the self-pain control stage, and less nausea and vomiting occur in patients with less pain; therefore, proper pain management is an essential component of the management of nausea and vomiting [32]. To prevent dizziness due to sudden body position changes, sudden blood flow movement was prevented, and dizziness due to low blood pressure was prevented by educating the patients and their guardians on the stepwise standing up method before early ambulation. Dangling, an interim stage of stepwise standing up, allowed the patients to move safely and prevented a dramatic reduction in blood pressure [33,34].

For anti-NVD education, a brochure and an explanation were offered to the patients, and an educational video was also shown using tablet PCs. Figure 1 shows part of the educational handbook on the stepwise standing up method, and Figure 2 shows the first screen of the video shown to the patients.

The details of the anti-NVD program were revised and supplemented by one nursing college professor, two head nurses, and two experienced nurses with over 10 years of work experience. The development of the program started on 1st August and ended on 31 October 2012.

### 2.4. Study Protocol

The experimental group received three sessions of anti-NVD education. First, the participants received a brochure on how to prevent postoperative nausea, vomiting, and dizziness and watched a relevant video using tablet PCs in the counseling room a day before their surgery. They also had time for a question-and-answer session with the researcher. Second, the nurse administered a painkiller when the patient arrived at the ward from the recovery room and offered education on deep breathing and adequate pain control after the patient was stabilized at a pain score of 3 points or below using a numerical rating scale (NRS). Third, a video on the stepwise standing up method was presented to the patient on the day after surgery before early ambulation, and then they received education again on the method of standing up. To the control group, existing educational materials were offered twice, on the day before surgery and postoperatively. Table 1 presents the details of the educational program offered to each group (Table 1).

### 2.5. Measurements

The participants’ general characteristics were surveyed through a questionnaire upon hospitalization, and their electronic medical records were referred to for operation-related characteristics.

#### 2.5.1. Nausea and Dizziness

Nausea and dizziness were measured twice: when the patients arrived at the ward after surgery and after they stood up for early ambulation on the day after surgery. An NRS was used to measure nausea and dizziness. When no nausea and dizziness existed, the score was 0, and a score of 10 denoted very severe nausea and dizziness.

#### 2.5.2. Vomiting and Frequency of Antiemetic Use

The number of vomiting events was measured, and the frequency of antiemetic use was surveyed immediately after surgery and 48 hours after surgery.

### 2.6. Data Collection

The data collection period of this study was from 1 January 2013 until 13 July 2013. To prevent dissemination of the experiment, 40 participants’ data were collected from 1 January 2013 until 10 April 2013 for the control group, and the data of 40 participants were collected from 11 April 2013 until 13 July 2013 for the experimental group. One participant in the control group who discontinued IV-PCA in the course of the research due to severe nausea and dizziness was excluded from the study; therefore, 39 and 40 participants from the control group and experimental group, respectively, participated in this study.

### 2.7. Ethics

This study received approval from the Institutional Review Board of Asan Medical Center (No. 2012-0662). The researcher individually interviewed patients hospitalized for surgery and explained the research goal and procedure. Prewritten consent was received only from those who stated their intention to participate in this study. It was explained that all data would only be used for research purposes, and they could discontinue participation during the experimental period. The confidentiality and anonymity of the collected data were maintained. A small gift was offered to the participants in return for their time and effort.

### 2.8. Data Analysis

The collected data were analyzed using SPSS version 18.0 (SPSS Inc., Chicago, IL, USA). The general and surgery-related characteristics of the participants were analyzed as mean values, standard deviation (SD) or variance, and percentages as descriptive statistics. The homogeneity of the two groups was verified using the chi-square test, and differences in nausea, vomiting, dizziness, and the frequency of antiemetic use were analyzed with the *t*-test.

## 3. Results

There were 79 participants (39 and 40 in the control and experimental groups, respectively). Homogeneity between groups was verified, without significant differences in general characteristics (sex, age, body mass index, education, religion, marital status, smoking, status of menopause, car sickness, and hyperemesis gravidarum), operation-related characteristics (surgical method, anesthesia duration, time elapsed from anesthesia to standing up, IV-PCA administration, ASA class, and the experience of hyperemesis gravidarum and general anesthesia), or the degree of nausea, vomiting, and dizziness experienced before arrival at the ward after surgery (Table 2).

Table 3 shows the results for nausea, vomiting, and dizziness immediately after standing up on the day after surgery and the frequency of antiemetic use from surgery until 48 hours after surgery. The scores for nausea immediately after standing up on the day after surgery were 0.68 and 1.95 points, respectively, in the experimental and control groups. The experimental group had a significantly lower level of nausea than the control group (t = 2.564, *p* = 0.013). The number of vomiting events in the control group on the day after surgery was two compared to zero in the experimental group, which was not a significant difference (OR = 0, *p* = 0.900). The dizziness scores of the experimental and control groups on the day after surgery were 1.45 and 3.51 points, respectively, which constituted a significant difference (t = 3.689, *p* < 0.001). The numbers of antiemetic use in the experimental and control groups were 0.42 and 1.49 times, respectively, which was a statistically significant difference (t = 2.238, *p* = 0.030).

## 4. Discussion

This study investigated the effects of an anti-NVD education program on reducing nausea, vomiting, and dizziness in a clinical trial targeting patients undergoing abdominal surgery using IV-PCA under general anesthesia. Nausea and dizziness in the experimental group, which received the education 30 minutes before surgery, 20 minutes after surgery, and stepwise standing up education 15 minutes before early ambulation, were significantly lower than in the control group. The frequency of antiemetic use in the experimental group was significantly lower than in the control group.

The findings of this study demonstrate the need for interventions to minimize nausea, vomiting, and dizziness in patients undergoing abdominal surgery under general anesthesia. The stepwise standing up method was confirmed to reduce nausea and dizziness when standing up. The finding that the anti-NVD program reduced nausea, dizziness, and the frequency of antiemetic use without the need for additional drugs or equipment has favorable implications from a cost-effectiveness standpoint.

The occurrence rates of nausea in the experimental and control groups immediately after standing up on the day after surgery were 17.5% and 48.7%, respectively. The occurrence of dizziness was much higher, as shown by rates of 38.0% and 77.0% in the experimental and control groups, respectively. A meaningful aspect of this study is that patients’ dizziness was confirmed in a clinical trial through objective data. The finding of a higher prevalence of dizziness than that of nausea and vomiting is conjectured to be related to the drug composition of the IV-PAC (ketarolac and fentanyl) used in this research. In a study of patients undergoing gynecological abdominal surgery using IV-PCA with ketarolac and fentanyl, the scores for postoperative discomfort were 3.29, 3.10, and 2.79 points for dizziness, tension, and nausea, respectively. The highest score was found for discomfort due to dizziness, and the need for interventions targeting dizziness was therefore inferred [35].

In studies of patients undergoing surgery for stomach cancer and colon cancer, the side effect of dizziness was found most frequently when IV-PCA using ketarolac, fentanyl, and ondansetron was applied [31]. If IV-PCA involves a combination of ketarolac and fentanyl, an active intervention for dizziness is necessary. Since dizziness was the reason for suspending IV-PCA in 93.0% of cases, interventions to reduce dizziness are essential for the purposes of both pain reduction and cost-effectiveness [36].

Vomiting after surgery occurred in two cases in the control group (5%), and in no cases in the experimental group. This finding of infrequent vomiting stands in sharp contrast with a previous study of patients undergoing gynecological surgery, which reported that vomiting occurred in 25% of the experimental group and 76% of the control group, respectively. An explanation for this discrepancy may be that the previous study included women in a high-risk group for nausea and vomiting [37]. The present study provides primary data for strategies to reduce nausea and dizziness upon standing up on the day after surgery. These findings will inform nursing interventions aiming at reducing postoperative nausea and dizziness.

Since the participants of this study were limited to patients hospitalized in the urology department in a general hospital who underwent nephrectomy, care should be taken when generalizing the research results. Further research is needed to confirm the effects and usefulness of the anti-NVD education program.

## 5. Conclusions

The effects of an anti-NVD education program before surgery on postoperative nausea, vomiting, and dizziness in patients undergoing abdominal surgery were evaluated. The anti-NVD education program reduced postoperative nausea and dizziness, as well as the frequency of antiemetic use. Stepwise standing up education after surgery using a video proved to be an effective intervention for easing nausea and dizziness immediately after standing up from bed.

## Figures and Tables

**Figure 1 ijerph-18-06124-f001:**
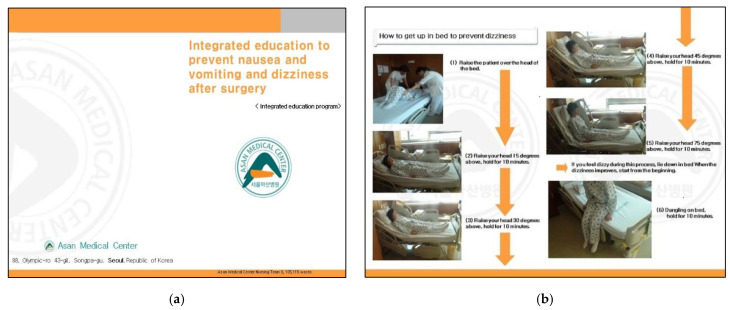
Anticipatory integrated education program brochure. (**a**) The cover of the educational program for nausea, vomiting, and dizziness prevention, and (**b**) photographs depicting the stepwise standing up method. Step 1, move the patient to the top part of the bed. Step 2, raise the bed by 15° and maintain that position until dizziness disappears. Steps 3 to 5, raise the bed by 30°, 45°, and 75° and maintain the position for 10 minutes each until dizziness disappears. In the last step, induce the return of venous blood by moving the patient’s legs while he or she sits on the bed. If no dizziness is felt, carry out early ambulation by standing up with a caregiver. If dizziness occurs during early ambulation, lie down on the bed. If dizziness disappears, try to stand up step-by-step from the start again. Reproduced with permission from EES, YHS, HJP, ESL.

**Figure 2 ijerph-18-06124-f002:**
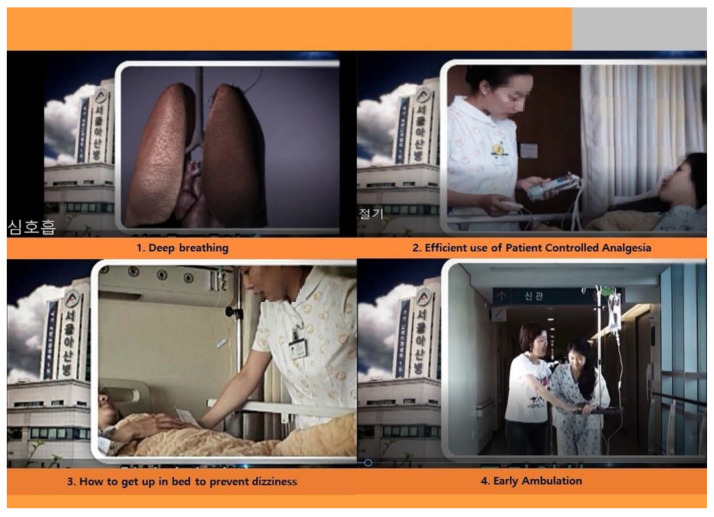
Anticipatory integrated education program (video screen capture). Video screen capture of footage on deep breathing, inspirometer use, intravenous patent-controlled analgesia use, the stepwise standing up method, and early ambulation. Reproduced with permission from EES, YHS, HJP, ESL.

**Table 1 ijerph-18-06124-t001:** Educational content for the experimental group and control group.

Item	Experimental Group	Control Group
Content	Factors causing postoperative nausea, dizziness, and vomiting Deep breathing Effectively using the IV-PCA How to get up in bed to prevent dizziness The need for early ambulation to prevent paralytic ileus	Deep breathing Effectively using the IV-PCA
Method	Booklet Tablet PC (video) 1:1 Face-to-face education	Brochure 1:1 Face-to-face education
No. and timing of education sessions	3 times Evening on the day before surgery (30 min) Operative day: upon arrival at the ward (20 min) The next day (before standing up; 15 min)	2 times Evening on the day before surgery (15 min) Operative day: upon arrival at the ward (15 min)

IV-PCA, intravenous patient-controlled analgesia.

**Table 2 ijerph-18-06124-t002:** Homogeneity tests between the two groups at baseline (*n* = 79).

Variable	Category	Experimental (*n* = 40) No. (%) or Mean ± SD	Control (*n* = 39) No. (%) or Mean ± SD	t/*X*^2^	*p*
Sex	Male	30 (75.0)	24 (61.5)	1.654	0.20
	Female	10 (25.0)	15 (38.5)		
Age (yr)		48.28 ± 13.64	49.00 ± 14.56	0.228	0.82
BMI (kg/m^2^)		24.83 ± 2.73	24.53 ± 3.31	−0.436	0.67
Eduction	High school	21 (52.5)	20 (51.3)	6.504	0.26
	Above college	19 (47.5)	19 (48.7)		
Religion	Yes	19 (47.5)	22 (56.4)	0.628	0.42
	No	21 (52.5)	17 (43.6)		
Married	Yes	33 (82.5)	35 (89.7)	1.821	0.40
	No	7 (17.5)	4 (10.3)		
Smoking	Yes	13 (32.5)	11 (28.2)	1.154	0.56
	No	27 (67.5)	28 (71.8)		
Menopause	Yes	3 (30.0)	7 (46.7)	0.694 ^1^	0.40
	No	7 (70.0)	8 (53.3)		
Carsickness	Yes No	2 (5.0) 38 (95.0)	7 (17.9) 32 (82.1)	3.524 ^1^	0.26
Hyperemesis	Yes No	4 (50.0) 4 (50.0)	11 (73.3) 4 (26.7)	1.252 ^1^	0.37
Operation	Open	3 (7.5)	5 (12.8)	0.614 ^1^	0.48
	Laparotomy	37 (92.5)	34 (87.2)		
Anesthesia time (min)		198.20 ± 44.50	212.28 ± 43.97	1.353	0.18
Interval from anesthesia start to standing (min)		1180.85 ± 193.74	1144.64 ± 206.95	−0.803	0.42
IV-PCA (mixed ondansetron)	Yes	17 (42.5)	13 (33.3)	0.704	0.40
	No	23 (57.5)	26 (66.7)		
ASA class	Class 1	18 (45.0)	23 (59.0)	1.545	0.21
	Class 2	22 (55.0)	16 (41.0)		
Previous nausea experience	Yes	3 (5.0)	2 (5.1)	0.001 ^1^	1.00
	No	38 (95.0)	37 (94.9)		
Previous anesthesia experience	Yes	14 (35.0)	20 (51.3)	2.135	0.14
	No	26 (65.0)	19 (48.7)		
Nausea		10 (25.0), 0.90 ± 1.91	11 (28.2), 1.21 ± 2.36	0.632	0.53
Dizziness		6 (15.0), 0.60 ± 1.70	9 (23.0), 0.77 ± 1.81	0.427	0.67
Vomiting		0	0		/

Values are presented as number (%) or mean ± standard deviation. BMI, body mass index; ASA, American Society of Anesthesiology; IV-PCA, intravenous patient-controlled analgesia. ^1^ Fisher exact test.

**Table 3 ijerph-18-06124-t003:** Comparison of levels of postoperative nausea, vomiting, dizziness and the frequency of antiemetic use after education (*n* = 79).

Group	N (%)	Mean ± SD	t, OR	*p*
Nausea			2.564	0.013
Experimental (*n* = 40)	7 (17.5)	0.68 ± 1.61		
Control (*n* = 39)	19 (48.7)	1.95 ± 2.67		
Vomiting			0	0.900
Experimental (*n* = 40)	0	-		
Control (*n* = 39)	2 (5.0)	0.05 ± 0.22		
Dizziness			3.689	<0.001
Experimental (*n* = 40)	15 (38.0)	1.45 ± 2.06		
Control (*n* = 39)	30 (77.0)	3.51 ± 2.84		
Antiemetic use (from arrival at the ward until 48 hr)			2.238	0.030
Experimental (n = 40)	-	0.42 ± 0.93		
Control (*n* = 39)	-	1.49 ± 2.81		

SD: standard deviation, OR: odd ratio.

## Data Availability

Data are available upon request.

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
