# Peer review of "Effectiveness of Integrated Education to Reduce Postoperative Nausea, Vomiting, and Dizziness after Abdominal Surgery under General Anesthesia"

_ijerph, 2021, doi:10.3390/ijerph18116124_

Round 1

Reviewer 1 Report

It is my pleasure to review this interesting study. The theme is novel and interesting. This research provides a scope to understand the intervention for complication of abdominal surgery. I had some suggestions to improve the paper.

Previous studies that have provided intervention for surgical complications should be presented. In particular, why do you choose education as an intervention method? What is the difference between this study and previous study?

It need the reference of g*power program. According to URL (https://www.psycholo-122 gie.hhu.de/arbeitsgruppen/allgemeine-psychologie-und-arbeitspsychologie/gpower), one or both of the following references could be included.
1) Faul, F., Erdfelder, E., Lang, A.-G., & Buchner, A. (2007). G*Power 3: A flexible statistical power analysis program for the social, behavioral, and biomedical sciences. Behavior Research Methods, 39, 175-​191.
2) Faul, F., Erdfelder, E., Buchner, A., & Lang, A.-G. (2009). Statistical power analyses using G*Power 3.1: Tests for correlation and regression analyses. Behavior Research Methods, 41, 1149-​1160.

The results of this study do not refer to a recent moment, but almost 8 years ago.

Table 3, What does mean “N (%)”? Is it the frequency of nausea, vomiting, or dizziness? Or, is it the number of subject who have experienced nausea, vomiting, or dizziness? In line 266-267, this data was considered frequency.

Reviewer 2 Report

Easy to understand.

Method simply discribed and adequated.

It would be better to have table 3 before the point 4 (discussion)

Author Response

Comments: Comments and Suggestions for Authors: Easy to understand. Method simply described and adequated. -->It would be better to have table 3 before the point 4 (discussion)

  • Response: Thank you for your comments. Table 3 is moved to before discussion session.

Reviewer 3 Report

Revision of the article:

Effectiveness of integrated education to reduce postoperative nausea, vomiting, and dizziness after adbominal surgery under general anesthesia

This article aims to study the effectiveness of an anti-NVD (nausea, vomiting and dizziness) education program to solve problems such as nausea, vomiting and dizziness in a sample of 79 participants (39 participants in the control group and 40 in the experimental group).

This quasi-experimental study is well designed and developed. Some limitations are the small sample size and having reduced the sample only to people who have undergone a nephrectomy, which makes it difficult to generalize the results

Even so, in my opinion, and although a few years have passed since the study was carried out, the results are publishable since they can serve as a reference for other studies that want to contribute knowledge in the same line of research.

Kind regards

Author Response

Comments: Effectiveness of integrated education to reduce postoperative nausea, vomiting, and dizziness after abdominal surgery under general anesthesia. This article aims to study the effectiveness of an anti-NVD (nausea, vomiting and dizziness) education program to solve problems such as nausea, vomiting and dizziness in a sample of 79 participants (39 participants in the control group and 40 in the experimental group). This quasi-experimental study is well designed and developed. Some limitations are the small sample size and having reduced the sample only to people who have undergone a nephrectomy, which makes it difficult to generalize the results. Even so, in my opinion, and although a few years have passed since the study was carried out, the results are publishable since they can serve as a reference for other studies that want to contribute knowledge in the same line of research. 

  • Response: Thank you so much for your complimentary advices.

Round 2

Reviewer 1 Report

The authors have improved the manuscript and have adequately responded to the majority of my comments.